# Metabolomics by NMR Combined with Machine Learning to Predict Neoadjuvant Chemotherapy Response for Breast Cancer

**DOI:** 10.3390/cancers14205055

**Published:** 2022-10-15

**Authors:** Marcella R. Cardoso, Alex Ap. Rosini Silva, Maria Cecília R. Talarico, Pedro H. Godoy Sanches, Maurício L. Sforça, Silvana A. Rocco, Luciana M. Rezende, Melissa Quintero, Tassia B. B. C. Costa, Laís R. Viana, Rafael R. Canevarolo, Amanda C. Ferracini, Susana Ramalho, Junier Marrero Gutierrez, Fernando Guimarães, Ljubica Tasic, Alessandra Tata, Luís O. Sarian, Leo L. Cheng, Andreia M. Porcari, Sophie F. M. Derchain

**Affiliations:** 1Department of Obstetrics and Gynecology, Division of Gynecologic and Breast Oncology, School of Medical Sciences, University of Campinas (UNICAMP–Universidade Estadual de Campinas), Campinas 13083881, SP, Brazil; 2Department of Pathology, Massachusetts General Hospital, Harvard Medical School, Boston, MA 02129, USA; 3MS4Life Laboratory of Mass Spectrometry, Health Sciences Postgraduate Program, São Francisco University, Bragança Paulista 12916900, SP, Brazil; 4Brazilian Biosciences National Laboratory (LNBio), Brazilian Center for Research in Energy and Materials (CNPEM), Campinas 13083970, SP, Brazil; 5Department of Medical Genetics, School of Medical Sciences, University of Campinas (UNICAMP–Universidade Estadual de Campinas), Campinas 13083887, SP, Brazil; 6Laboratory of Chemical Biology, Department of Organic Chemistry, Institute of Chemistry, University of Campinas (UNICAMP–Universidade Estadual de Campinas), Campinas 13083862, SP, Brazil; 7Laboratory of Nutrition and Cancer, Department of Structural and Functional Biology, Biology Institute, University of Campinas (UNICAMP–Universidade Estadual de Campinas), Campinas 13083862, SP, Brazil; 8Department of Cancer Physiology, H. Lee Moffitt Cancer Center and Research Institute, Tampa, FL 33612, USA; 9Postgraduate Program in Medical Sciences, School of Medical Sciences, University of Campinas (UNICAMP–Universidade Estadual de Campinas), Campinas 13083970, SP, Brazil; 10Women’s Hospital Prof. Dr. José Aristodemo Pinotti, Centro de Atenção Integral à Saúde da Mulher (CAISM), University of Campinas (UNICAMP–Universidade Estadual de Campinas), Campinas 13083891, SP, Brazil; 11Laboratorio di Chimica Sperimentale, Istituto Zooprofilattico Sperimentale delle Venezie (IZSVe), Viale Fiume 78, 36100 Vicenza, Italy; 12Department of Radiology, Massachusetts General Hospital, Harvard Medical School, Boston, MA 02129, USA

**Keywords:** 1H-NMR, breast neoplasms, magnetic resonance spectroscopy, metabolism, untargeted metabolomics, drug resistance

## Abstract

**Simple Summary:**

Neoadjuvant chemotherapy (NACT) is offered to breast cancer (BC) patients to downstage the disease. However, some patients may not respond to NACT, being resistant. We used the serum metabolic profile by Nuclear Magnetic Resonance (NMR) combined with disease characteristics to differentiate between sensitive and resistant BC patients. We obtained accuracy above 80% for the response prediction and showcased how NMR can substantially enhance the prediction of response to NACT.

**Abstract:**

Neoadjuvant chemotherapy (NACT) is offered to patients with operable or inoperable breast cancer (BC) to downstage the disease. Clinical responses to NACT may vary depending on a few known clinical and biological features, but the diversity of responses to NACT is not fully understood. In this study, 80 women had their metabolite profiles of pre-treatment sera analyzed for potential NACT response biomarker candidates in combination with immunohistochemical parameters using Nuclear Magnetic Resonance (NMR). Sixty-four percent of the patients were resistant to chemotherapy. NMR, hormonal receptors (HR), human epidermal growth factor receptor 2 (HER2), and the nuclear protein Ki67 were combined through machine learning (ML) to predict the response to NACT. Metabolites such as leucine, formate, valine, and proline, along with hormone receptor status, were discriminants of response to NACT. The glyoxylate and dicarboxylate metabolism was found to be involved in the resistance to NACT. We obtained an accuracy in excess of 80% for the prediction of response to NACT combining metabolomic and tumor profile data. Our results suggest that NMR data can substantially enhance the prediction of response to NACT when used in combination with already known response prediction factors.

## 1. Introduction

Breast cancer (BC) is the most common cancer worldwide [1] and the leading cause of cancer-related mortality among women in Brazil [2]. Although BC presents histological similarity, clinically it can be very heterogeneous, with several phenotypic and genotypic subtypes [3], which results in a challenge in treatment effectiveness. Molecular and immunohistochemical markers generally are used to classify BC subtypes. The classification is based on the expression of hormonal receptors (HR) such as estrogen and progesterone receptors (ER; PR), human epidermal growth factor receptor 2 (HER2), and the cell proliferation marker (nuclear protein Ki67). These markers are also related to therapeutic decisions and prognostics [4,5].

One of the treatments used prior to definitive surgical therapy is neoadjuvant chemotherapy (NACT), which has gained attention due to its ability to reduce tumor size and cancer burden, avoiding mutilating surgical procedures [6]. It also has the potential for increasing resectability and controlling the micrometastatic disease. NACT provides a viable alternative when there is poor radiotherapy access or there are unavoidable delays in delivering radiotherapeutic treatment. To evaluate whether the patient had a favorable response after completion of NACT, pathological complete response (pCR) is defined as the absence of invasive tumors in the breast and axilla. This response is associated with improved long-term survival rates or far lower risk of subsequent recurrence [7]. BC patients who achieved a pCR after the NACT showed higher rates of disease-free survival (DFS) and overall survival (OS) than women with residual disease in their surgical specimens obtained after NACT [8]. Unfortunately, about 30% of patients show a pCR to NACT.

Patients with HR-positive tumors have a lower probability of pCR after NACT than patients with HER2-positive and triple-negative tumors [9]. The well-known HR, HER2, as well as the Ki67 markers triad, explain a substantial portion of the NACT response variability. Nevertheless, a better understanding of the mechanisms that define the response to NACT is still needed to determine which patients will gain clinical benefits from NACT. Many patients undergoing NACT have the adverse effects of therapy without enjoying pCR, and oncologists are largely incapable of predicting unsuccessful outcomes. To further improve the knowledge of the molecular events leading to pCR after NACT, we designed and carried out the present metabolomic study to construct a biomolecular and metabolic portrait of the biochemical processes in fluids and tissues of BC patients who had undergone NACT [10,11,12].

Metabolites are final byproducts acquired from the interaction between intracellular pathways and their microenvironment [13]. Metabolomics, when faced with a stimulus such as a disease, measure a comprehensive set of metabolites, thus representing the bioactivity of a system [14]. Thus, in cancer, metabolomics detects oncological developments by evaluating measurable metabolic profiles and selecting metabolic pathways through global variations in metabolites [15,16]. Specifically for breast cancer, several articles have proposed that assessing a metabolite profile might allow for an understanding of the biochemical processes that occurred or were occurring at the time of diagnosis [17,18,19]. In addition, in the chemoresistance field, metabolomics allows for the individual characterization of the patient, enabling the personalization of treatment with strategies focused on maximizing the action of drugs. Thus, metabolomics can be developed as a sensitive prognostic tool for associated therapy for diseases such as breast cancer [20,21]. However, a few studies have evaluated metabolites associated with better response to NACT using metabolomics [22,23,24].

The use of bioinformatics allows the use of approaches for predictive modeling, using medical data, for cancer evaluation [25]. For example, machine learning is able to generate models for prediction by testing vastly through model and parameter space, as opposed to traditional statistics approaches, which have a limited set of hypotheses [26,27,28,29,30].

In our study, we analyzed the serum metabolites of patients with different molecular subtypes of BC who had undergone NACT using Nuclear Magnetic Resonance (NMR). We also created machine learning classifiers by correlating the observed metabolites with the expression of BC markers and creating models to predict the response to NACT. These models were able to predict the response to NACT using pre-treatment serum samples.

## 2. Materials and Methods

### 2.1. Subjects

This is a prospective cohort study on 80 women aged between 29 and 77 years with invasive breast carcinoma who underwent NACT followed by surgery. Participants were diagnosed and treated at the Women’s Hospital (Hospital da Mulher Prof. Dr. José Aristodemo Pinotti, Centro de Atenção Integral à Saúde da Mulher–CAISM) of the University of Campinas (UNICAMP) in Brazil between January 2017 and January 2019. Demographic and clinical data from patients’ records are summarized in Table 1. The biological samples were stored in the CAISM’s biobank (CONEP 56, Campinas, SP, Brazil). All study subjects signed informed consent forms before removing their biological samples and being included in the institutional biobank. The Research Ethics Committee approved the study of UNICAMP (CAAE, 69699717.0.0000.5404).

#### 2.1.1. Clinical and Histopathologic Diagnosis of Breast Cancer

Tissue samples were collected with an ultrasound-guided percutaneous needle (core) biopsy either at biopsy (pre-treatment specimens) or during surgical resection after NACT (post-treatment specimens). The samples were formalin-fixed and paraffin-embedded. Hematoxylin-eosin-stained sections were reviewed to confirm the histologic diagnosis. NACT was prescribed according to the standard protocols, including doxorubicin and cyclophosphamide, followed by paclitaxel (with carboplatin in triple-negative cases). HER2-positive patients received trastuzumab. See Appendix A for the detailed information of the therapeutic regimen. After NACT, all women underwent surgical treatment (mastectomy or quadrantectomy with sentinel lymph node biopsy or axillary lymph node dissection). All histological diagnoses were determined according to the World Health Organization (WHO) criteria and following the grade as per the Nottingham classification (Appendix A) [31].

#### 2.1.2. Immunohistochemical Diagnosis

For BC subtype classification, a conventional manual immunohistochemical technique was used [31,32,33,34,35]. Anti-estrogen receptor (ER, clone 1D5, diluted at 1:1000 *v*/*v*), anti-progesterone receptor (PR, clone PR636, diluted at 1:800 *v*/*v*), anti-HER2 (Clone PN2A, diluted at 1:1100 *v*/*v*), and anti-Ki67 (clone MIB1, diluted at 1:500 *v*/*v*) were used as primary antibodies (all by Dako, Agilent, Santa Clara, CA, USA). Evaluation of ER and PR was performed accordingly, on the diagnostic (pre-treatment) specimens [32]. Cases were considered positive when ≥1% of tumor cells were positive. For Ki67, we scored the percentage of positive tumor cells in a minimum of 500 cells in randomly selected representative fields [33]. When the mean percentage of stained cells was higher than the median (40%), cases were assigned as Ki67 high. Human epidermal growth factor receptor 2 (HER2) staining was scored as 0+/1+ (negative), 2+ (equivocal), or 3+ (positive). Equivocal (2+) cases were further confirmed by in situ hybridization, according to the recommendations of the American Society of Clinical Oncology/College of American Pathologists (ASCO/CAP) [34]. Due to tumor heterogeneity, hormone-receptor-negative cases and/or HER2-negative had the immunohistochemistry repeated in the surgical specimen of women with a residual disease for subtype confirmation [35].

#### 2.1.3. Response to NACT

The response to NACT was evaluated on surgical specimens removed after NACT according to the Residual Cancer Burden (RCB) guidelines [36,37]. For statistical analyses, all cases were clustered into two major groups based on the response to NACT: (a) cases with a pCR and RCB-I (minimum residual disease) as sensitive, and (b) cases with residual breast carcinoma RCB-II (moderate residual disease) and RCB-III (extensive residual disease) as resistant [38].

### 2.2. Untargeted Nuclear Magnetic Resonance (NMR) Metabolomic Analysis of Serum Samples

Aliquots of peripheral blood were collected in dry tubes (Vacuette^®^ tube 2.5 mL CAT Serum Separator Clot Activator 13 × 75 red cap-white ring, non-ridged, North Carolina, USA) and obtained from women before they initiated the NACT. Immediately after blood collection, the tubes were mixed by inverting several times and incubated for ≥30 min at 4 °C. The tubes were centrifuged for 10 min at 1000× *g*. The samples were stored at −80 °C until analysis.

Serum samples were thawed at room temperature before the spectroscopy analysis. Then, 400 μL of serum was slowly mixed with 200 μL of D_2_O (99.9% deuterium oxide with 0.03% of TSP) and transferred to 5 mm NMR tubes. The proton NMR spectra (1H-NMR) were acquired at 298 K using a Varian Inova^®^ of 599.887 MHz NMR spectrometer (Agilent Technologies^®^ Inc., Santa Clara, CA, USA) equipped with a triple resonance cryoprobe.

Regular one-dimensional 1H-NMR spectra were obtained using CPMG (Carr–Purcell–Meiboom–Gill) pulse sequence with ns = 128 with 8 K data points at a spectral sweep width of 16 ppm; total relaxation delay of 4 s; water presaturation applied during 2 s delay; T2 filtering was obtained with an echo time of 300 μs repeated 166 times, resulting in a total duration of effective echo time of 50 ms. The NMR acquisitions were performed at the Brazilian Biosciences National Laboratory (Brazilian Center for Research in Energy and Materials, CNPEM, Campinas, SP, Brazil).

#### 2.2.1. NMR Analysis

The obtained spectra were processed using the MestreNova software (MestrelabResearch S.L.). The chemical shifts of the spectra were referenced using the doublet corresponding to the lactate’s methyl group (–CH_3_) at 1.324 ppm (3H, d, 3JHH = 7.0 Hz). The extent of information was reduced by using a binning of 0.005 ppm. The samples were normalized by the sum. The processed spectra were converted into data matrices to prepare them for identification and analysis. In addition, regions of the spectrum corresponding to substances with a potential for interference in the analyses, such as water (4.40–5.23 ppm), were removed. The NMR analyses were performed at the Biological Chemistry Laboratory from UNICAMP. Chenomx NMR^®^ Suite 8.1 software (Chenomx^®^ Inc., Edmonton, AB, Canada) was used to quantify relative concentrations.

#### 2.2.2. Statistical Analysis

All statistical calculations were performed using the R Foundation for Statistical Computing (v3.6.2), Vienna, Austria. The epitools package was used for the odds ratio (OR) calculation of clinical characteristics of the women according to their response to NACT. Confidence levels of 95% and *p*-values <0.05 were assumed as statistically significant [39]. In addition, we used standard methodology to calculate the sample size for the study. Assuming a 0.05 significance level, a Cohen coefficient of 0.8 (high), the power for our analysis sits at a comfortable 0.801, considering the proportion of NACT sensitive/resistant patients in the cohort of 80 women with complete clinical data available for analyses. The caret package was used for recursive feature elimination (RFE) and logistic regression (LR) modeling [40]. RFE was applied for continuous elimination of features (i.e., metabolites or clinical markers) with a low contribution to the model [41,42,43].

Nine classification prediction models for the response to NACT were generated based on the alternative combinations of HR, Ki67, HER2 statuses, and the metabolite panel (Appendix A). We performed separate analyses for the entire (Appendix A) and triple-negative/HER2+ tumors (*n* = 45, Appendix A).

Models were built based on a training set comprising 75% of the data. To evaluate the performance of the models, we performed Leave-One-Out Cross-Validation (LOOCV). The area under the curve (AUC) of the receiver operator characteristics (ROC) curve of the selected features was also used to evaluate their prediction power. A validation set comprising 25% of data was also used for performance evaluation in terms of sensitivity, specificity, and accuracy [44].

For metabolomics analysis, normalization by sum and pareto scaling data were performed using the web platform MetaboAnalyst™ 5.0 [45]. Using this platform, we further interrogated the KEGG Library for quantitative Metabolite Pool enrichment, and the pathways with a *p*-value < 0.05 were considered significant.

## 3. Results

### 3.1. Subjects and Clinical Features

Eighty patients included in this study and undergoing NACT presented invasive ductal carcinoma (100%), most of which were histological grade 3 (51%), ER-positive (71%), and PR-positive (65%). Patients with tumors of histological grade 3 had a higher probability of pCR/RCB-I (OR = 5.57 (1.439–21.470); *p* = 0.0161) than their counterparts whose tumors were grade 1/2. In contrast, women with HR-positive tumors were less likely to enjoy pCR/RCB-I (OR = 0.18 (0.04–0.670); *p* = 0.005) than the women with non-luminal tumors. Table 2 shows the distribution of women according to the disease characteristics and NACT response.

### 3.2. NMR-Based Metabolomic Analysis

Untargeted NMR analysis identified and quantified the relative abundances of more than 27 compounds, including 14 amino acids (arginine, asparagine, glutamate, histidine, leucine, lysine, phenylalanine, proline, serine, threonine, tyrosine, and valine) and other metabolites (Appendix A). Overall, no significant differences in the number of metabolites identified by 1H-NMR in the spectra of resistant and sensitive patients were observed. Representative spectra obtained from serum patients in the resistant and sensitive groups are shown in Figure 1. Univariate analysis shows that leucine and formate were found to be significantly altered when comparing resistant and sensitive women (*p*-value < 0.05) (Figure 2 and Appendix A); however, their predictive power evaluated using a logistic regression model showed to be unsatisfactory (AUC = 0.67) (data not shown).

### 3.3. Classification Models for Predicting Response to NACT

Logistic regression (LR) models have supported the assessment of biomarkers in cancer [46,47,48,49] and can also be coupled with RFE [50], as we present in Figure 3. When compared to the univariate approach, we observed an increase in the predictive power for the model using only the abundance of the metabolites (AUC = 0.83 – Model I, Figure 3A). Using the LR-RFE, we found formate, proline, valine, and leucine as potential markers of NACT response (Figure 3B). By combining the abundances of these metabolites with the information on HR status (Model II), HR and HER2 (Model VI), and Ki67, HER, and HR (Model VIII), we obtained AUC values higher than those obtained for the model that solely used metabolites.

The contributions of the metabolites were found to be more significant for the models than the contribution of the molecular markers, as presented by the observed coefficient values (Figure 3B). For the total contribution of the HR, Ki67, HER2, and serum metabolites as predictors of the response to NACT in the different models obtained, see Appendix A.

The coefficient values display the patient’s chance of being resistant to NACT. A positive value means a positive correlation with NACT resistance, whereas a negative value means a positive correlation with NACT sensitivity. For example, formate, proline, valine, HR+, and HER2− display positive coefficient values, thus being directly related to the NACT resistance. It means that once any of these predictors show an increase, the patient’s chance of being resistant to NACT increases. Conversely, leucine, HR−, and HER2+ have negative coefficient values and are inversely related to NACT resistance or directly associated with NACT sensitivity. It means that an increase in leucine concentration, keeping the remaining predictors constant, increases the patient’s chance of being sensitive to the treatment, in accordance with what was previously pointed out by the univariate analysis of the metabolite set. Ki67, both positive and negative, was found to have a low influence on the models that contain it.

When exploring the performances of the models based on their predictive power and agreement with the pathologic results, equivalent specificity rates were found for the eight models when considering the training set (Appendix A). For the validation sets, the specificity reached 75% only for the models II and V (Figure 3C), which were considered the ones with the highest efficiency in the classification. The sensitivity of the models, for both training and validation sets, was higher than 70% for all the models (Appendix A). The results suggest that the interconnection of clinical variables (HR/Ki67/HER2) and serum metabolites can improve the prediction of the response to NACT. When the analyses were performed in a restricted cohort of patients with triple-negative and HER2+ tumors, similar performance estimators were obtained (Appendix A).

To gain insights into metabolome changes associated with resistance and sensitivity to NACT, the pathway enrichment analysis was carried out with the metabolites found as discriminants for models I–VIII. Glyoxylate and dicarboxylate metabolism was the most enrolled pathway for acquiring resistance (Appendix A).

## 4. Discussion

The potential of the combination of HR, Ki67, and HER2 statuses and serum metabolites in predicting the response to NACT was evaluated in this study. The untargeted NMR analysis identified and quantified the relative abundances of 28 compounds, including 14 amino acids, carnitine, and other metabolites. Amino acids are an indispensable source of nutrients for all types of cells. Both essential and non-essential amino acids have an important role in providing build blocks for cell growth and proliferation. The critical role of these amino acids is even more important for tumor cells due to their rapid growth and proliferation, supporting protein synthesis [51]. It is also known that amino acids cannot cross the cell membrane without the assistance of specific transporters due to their hydrophilic component [52]. Therefore, since tumor cells require a high demand of amino acids to satisfy their rapid growth and proliferation, amino acid transporters’ expression is higher than normal cells [53]. Furthermore, the amino acid transporters are expressed in different levels in types of cancers and exhibit different properties in substrate selectivity [54].

When studying the serum abundance of this set of metabolites, we noticed the significant differences between the concentrations of leucine and formate. Leucine, which has been extensively studied for its role in breast cancer, showed a lower concentration in the serum of resistant women [55,56,57]. Saito et al., 2019, proposed that resistance is based on the increased expression of transporters that incorporate leucine to fuel the accelerated proliferation of cells resistant to therapy, which supports our observation of a lower concentration of this amino acid in resistant patients to NACT [58]. The higher concentration of leucine observed in the serum from sensitive in comparison to resistant patients could reflect less leucine uptake from sensitive tumors. One possible explanation could be a lower expression of L-type amino acids transporter 1 (LAT1) in sensitive tumors. Leucine is an essential amino acid [59] that activates the mammalian target of rapamycin (mTOR) which regulates cell growth and cell cycle progression [60], stimulating insulin signaling [61] or oxidized for energy purposes by tumors [62]. Therefore, within a lower leucine uptake, the mTOR pathway will be down-regulated, growth stimuli and energy source will be decreased, and consequently, less cell growth and proliferation could lead to the sensitive phenotype. Contrarily, Yang and collaborators (2018) performed a metabolomics study evaluating the plasma profile associated with response to neoadjuvant chemotherapy for colorectal cancer patients and found that leucine concentration was decreased in responsive patients [61]. Compared to Yang et al. (2018), this discrepancy in the present study could be explained by the heterogeneity among different cancers, such as cell type and histologic classification. A study by Saito and colleagues (2019) demonstrated that leucine imported by LAT1 was involved in tamoxifen resistance in ER-positive breast cancer [58].

In addition to leucine, another amino acid, arginine, had different serum profiles when comparing sensitive and resistant patients. Arginine is a semi-essential amino acid that could be derived primarily from diet and synthesis in the kidney [63]. Arginine is crucial in many biologic processes, including the immune system. Furthermore, arginine is essential to cellular growth and may become limited in cases of rapid proliferation such as malignancy [64]. Jayant and Anant (2017) evaluated the serum levels of arginine in breast cancer patients and found lower levels in cancer patients, independent of stage, compared to healthy controls [65]. In the present study, we found higher levels of arginine in resistant patients in comparison to sensitive ones. This result could reflect that the required arginine may differ between sensitive and resistant tumors.

In addition to alterations in the serum amino acid profile, other metabolites such as formate have their profile changed among sensitive and resistant breast cancer patients. Formate is produced by many metabolic pathways such as serine catabolism, through the mitochondrial pathway. Sterol synthesis and tryptophan catabolism are also involved in this process. Formate may have different directions depending on cell type and environmental conditions. In proliferating cells, such as cancer, formate contributes to the one-carbon (1C) demand for the synthesis of nucleotides [66]. The high demand for nucleotide, RNA, and DNA is essential for rapid tumor growth [67]. For this reason, the circulating formate levels are reduced in some cancer patients, such as breast and lung cancer, relative to healthy controls [68]. Formate was increased in resistant women and can be considered a potential predictive biomarker, in agreement with several publications on cancer [22,68,69]. Jiang et al. (2018) performed a pharmacometabolomics study in metastatic breast cancer patients evaluating the response to gemcitabine–carboplatin chemotherapy. The authors found that significantly lower baseline levels of serum formate in patients with the resistant disease may reflect the higher demand from them for alternate/additional nutritional sources to fuel the accelerated proliferation of breast cancer cells biologically more aggressive or resistant to therapy [70]. Contrarily, our results revealed significantly higher serum formate in resistant patients. This controversial result could be explained by the fact that the oxidative nature of cancers and the chemotherapy treatment for our patients may differ from those included in the study performed by Jiang.

Subsequently, we used the recursive feature elimination (RFE) method to assist in the selection of putative biomarkers. RFE continually removes the resources with low contribution scores based on the iterative method and then classifies each resource in each cycle to exclude the resources with low scores [43]. Studies have proposed that RFE allows the extraction of potential biomarker subsets among different cancer types [71,72,73]. Specifically, for breast cancer, RFE was applied to classify the complete pathological response and distinguish triple-negative breast cancer from other subtypes of breast cancer based on the selection of miRNA biomarkers [74,75]. Logistic regression (LR) models, following the RFE approach, are also widely used in classification problems applied to breast cancer [76,77,78,79].

In our study, RFE + LR was applied as a supervised learning machine by combining the abundance of metabolites with the clinical information on HR, Ki67, and HER2 status. The resulting classification models presented sensitivity and specificity values greater than 70% and 80%, respectively, for the training set. Remarkable performance was achieved for Model II, with 75% sensitivity and 83% specificity for the training set and 81% sensitivity and 75% specificity for the validation set. This indicates that combining the HR status with the metabolite panel (leucine/valine/formate/proline) can generate good classifiers of the response to NACT.

A few studies reported similar results for predicting the response to NACT in BC patients based on metabolite panels [22,23,24]. Lin et al. (2019) assessed the metabolic biomarker signature of serum samples of BC patients by liquid chromatography–mass spectrometry (LC-MS). The authors applied partial least squares discriminant analysis (PLS-DA) to build the statistical model of classification, which identified nine informative metabolites and achieved performances with a specificity of 100% and sensitivity of 81.2% [23]. Some authors recently examined serum metabolite levels during chemotherapy treatment. Debik et al. (2019) observed unfavorable changes in lipid levels during NACT. By using NMR, they observed no metabolic difference in serum samples from survivors and non-survivors, although a PLS-DA model based on their analysis of tissue achieved 72% accuracy in predicting a 5-year survival rate. Among other metabolites, lactate, glycine, choline, and alanine were reported as the critical variables useful for the model set-up [22]. Vignoli et al. (2020) also studied metabolic profiles capable of codifying a complete response to NACT within the ER-positive group. After investigating the plasma by NMR, they built a classifier with low performance. Among other findings, branched-chain amino acids, such as valine and isoleucine, were reported as the key differentiators of their groups of study [24]. Unlike the previous studies mentioned above, applying the RFE + LR method allowed us to obtain models with successful results in predicting the response to NACT that links the selected biomarkers with the clinical information of patients.

Glyoxylate and dicarboxylate metabolism was the pathway most impacted by the acquisition of resistance (Appendix A). This pathway has been associated with breast cancer cell metastasis by gas chromatography–mass spectrometry (GC-MS) and direct infusion mass spectrometry [80]. In addition, other pathways represented by our panel of metabolites are associated with breast cancer, such as the metabolism of branched-chain amino acids (BCAAs, i.e., valine, leucine, and isoleucine), the Aminoacyl-tRNA, arginine and proline, Pantothenate, and CoA [50,81]. Together, this set of relationships highlights the metabolic alterations involved in the cellular reprogramming that provoked the BC resistance to chemotherapy.

The current analysis is based on a strong dataset of comprehensive clinical and metabolomic data, derived from a well-maintained biobank, coupled with clinical facilities designed to collect world-class clinical data. We acknowledge, on the other hand, that a larger cohort of patients with non-luminal tumors would have enhanced the possibility of subset analyses aimed at refining the performance estimations on less common tumor types, such as triple-negative and HER2+ tumors. Although untreatable at this point, we are conducting further recruitment of patients with these less common tumor types.

## 5. Conclusions

In conclusion, NMR of serum offered rapid access to metabolic alterations in BC associated with resistance and sensitivity to NACT. The reported proof-of-principle study attested the power of these techniques in accelerating the definition of BC prognosis with excellent accuracy. Screening methods for predicting the response and effects of chemotherapies are much desired and would be of paramount importance for cancer patients. Despite BC being one of the most treatable cancers, a massive number of women do not benefit from NACT pre-treatment because of resistance to chemotherapy. Capitalizing on the close relationship between cancer formation and changes in serum, the analysis of the metabolites by NMR allowed insights into the metabolic alterations associated with marker status and resistance to NACT. When coupled by LR + RFE to the clinical marker status, a simple metabolomic panel can be applied successfully as an add-in prognosis and clinical forecasting for the response to NACT of BC patients. Although rigorous multisite validation of this untargeted approach with a larger patient pool is needed, this is the first milestone for developing an efficient strategy for the early discovery of NACT-resistant BC patients.

## Figures and Tables

**Figure 1 cancers-14-05055-f001:**
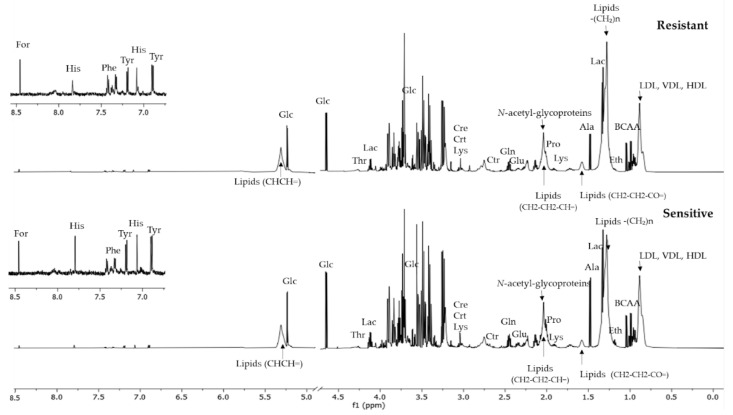
Representative 1H-NMR spectra from serum obtained from women before they initiated the NACT. Resistant and sensitive refer to response to neoadjuvant chemotherapy.

**Figure 2 cancers-14-05055-f002:**
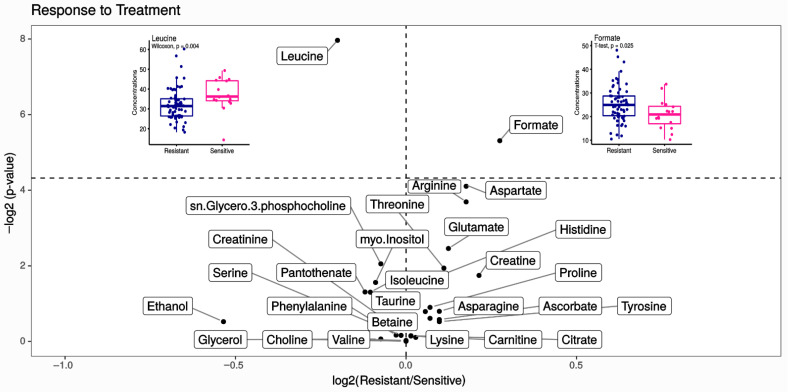
Volcano plot showing the metabolite variation by NACT outcomes (log2(resistant/sensitive)) in function of their statistical significance (log2(*p*-value)). Non-significant metabolites are plotted below the horizontal dashed line, and the metabolites above this line presented significant variation (*p*-value < 0.05). In addition, we show a boxplot for metabolites with significant variation (blue represents resistant patients and pink the sensitive ones); *p*-values were calculated using the Mann–Whitney–Wilcoxon test or *t*-test as a function of whether the data came from a normal distribution, proved with the Shapiro test.

**Figure 3 cancers-14-05055-f003:**
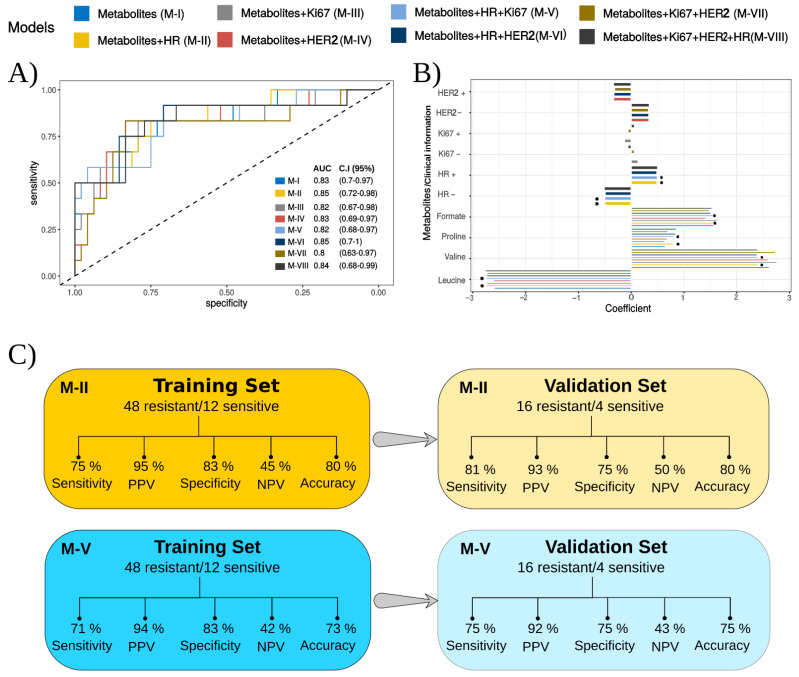
Classification models obtained with the combination of metabolites and clinical information. (**A**) The plot of the ROC curves for each model, containing the area under the curve (AUC). (**B**) The coefficients of the predictors and (**C**) performance of models II and V for training (cross-validation) and validation sets.

**Table 1 cancers-14-05055-t001:** Key patient information as related to response to neoadjuvant chemotherapy.

Characteristic		*n* (%)	Sensitive	Resistant	OR	*p*-Value
			*n* = 16 (%)	*n* = 64 (%)	(95% CI)	
Age	≥50	46 (57.5)	8 (50.0)	38 (59.4)	ref	
	<50	34 (42.5)	8 (50.0)	26 (40.6)	0.68 (0.23–2.05)	0.499
Race	Caucasian	69 (86.3)	15 (93.8)	54 (84.4)	ref	
	Non-Caucasian	11 (13.8)	1 (6.2)	10 (15.6)	2.78 (0.33–23.45)	0.293
Age of menarche	>12	42 (52.5)	9 (56.2)	33 (51.6)	ref	
	≤12	38 (47.5)	7 (43.8)	31 (48.4)	1.21 (0.4–3.64)	0.737
Menopause	No	36 (45.0)	9 (56.2)	27 (42.2)	ref	
	Yes	44 (55.0)	7 (43.8)	37 (57.8)	1.76 (0.58–5.32)	0.313
Hormone therapy	No	68 (85.0)	15 (93.8)	53 (82.8)	ref	
	Yes	12 (15.0)	1 (6.2)	11 (17.2)	3.11 (0.37–26.09)	0.233
Pregnancy (previous or current)	Yes	73 (91.3)	15 (93.8)	58 (90.6)	ref	
	No	7 (8.7)	1 (6.2)	6 (9.4)	1.55 (0.17–13.89)	0.681
Lactation *	Yes	63 (78.8)	13 (81.2)	50 (78.1)	ref	
	No	17 (21.2)	3 (18.8)	14 (21.9)	1.21 (0.3–4.86)	0.782
Smoking	Yes	17 (21.2)	4 (25.0)	13 (20.3)	ref	
	No	63 (78.8)	12 (75.0)	51 (79.7)	1.31 (0.36–4.73)	0.686
Chronic alcoholism	No	79 (98.8)	16 (100.0)	63 (98.4)	ref	
	Yes	1 (1.2)	0 (0.0)	1 (1.6)	NC	0.503
BMI	Normal weight	24 (30.0)	6 (37.5)	18 (28.1)	ref	
	Overweight	21 (26.3)	2 (12.5)	19 (29.7)	3.17 (0.56–17.77)	0.19
	Obese	35 (43.7)	8 (50.0)	27 (42.2)	1.13 (0.33–3.79)	0.849
Diabetes	No	71 (88.7)	15 (93.8)	56 (87.5)	ref	
	Yes	9 (11.3)	1 (6.2)	8 (12.5)	2.14 (0.25–18.5)	0.452
Family history of breast or ovarian cancer	No	59 (73.8)	9 (56.2)	50 (78.1)	ref	
	Yes	21 (26.2)	7 (43.8)	14 (21.9)	0.36 (0.11–1.14)	0.087

Sensitive (pCR and RCB I); resistant (RCB II and RCB III); NC, non-calculable; BMI, body mass index; * women who had no pregnancies. The following demographic and clinical data were obtained: age at diagnosis; self-declared ethnicity; age of menarche; menopausal status (premenopausal and postmenopausal); hormone therapy history; number of pregnancies; lactation regardless of the number of pregnancies or duration; smoking; chronic alcoholism; body mass index before the beginning of neoadjuvant chemotherapy (NACT) (normal weight: <25 kg/m^2^; overweight: 25 < 30; obese: ≥30); diabetes mellitus; family history of breast and/or ovarian cancer.

**Table 2 cancers-14-05055-t002:** Main breast cancer features as related to response to neoadjuvant chemotherapy.

Characteristic		*n* (%)	Sensitive	Resistant	OR	*p*-Value
			*n* = 16 (%)	*n* = 64 (%)	(95% CI)	
Histological grade	1/2	38 (48.75)	3 (18.8)	36 (56.2)	ref	
	3	41 (51.25)	13 (81.2)	28 (43.8)	0.18 (0.05–0.69)	0.006
Ki67	Low	34 (42.5)	5 (31.2)	29 (45.3)	ref	
	High	46 (57.5)	11 (68.8)	35 (54.7)	0.55 (0.17–1.76)	0.303
HER2	Negative	46 (57.5)	6 (37.5)	40 (62.5)	ref	
	Positive *	34 (42.5)	10 (62.5)	24 (37.5)	0.36 (0.12–1.12)	0.072
Tumor size **	T1/T2	53 (66.25)	12 (75.0)	41 (64.1)	ref	
	T3/T4	27 (33.75)	4 (25.0)	23 (35.9)	1.68 (0.49–5.82)	0.399
Regional lymph node	N0	33 (41.25)	6 (37.5)	27 (42.2)	ref	
	N1 or higher	47 (58.75)	10 (62.5)	37 (57.8)	0.82 (0.27–2.54)	0.732
Metastasis	M0	74 (92.5)	16 (100.0)	58 (90.6)	ref	
	M1	6 (7.5)	0 (0.0)	6 (9.4)	NC	0.094
Hormonal Receptor	Negative	21 (26.25)	9 (56.2)	12 (18.8)	ref	
	Positive	59 (73.75)	7 (43.8)	52 (81.2)	5.57 (1.73–17.96)	0.004

* Five patients had negative HER2 at core biopsy and positive HER2 in the surgical specimen; therefore, these patients were not treated with trastuzumab prior to surgery. ** One patient had an occult breast tumor with axillary disease. Legend: sensitive (pCR and RCB I); resistant (RCB II and RCB III); NC, non-calculable; Ki67, Ki67 protein (low ≤ 30 and high > 30); HER2, human epidermal growth factor receptor (scored as 0+/1+ (negative), 2+ (equivocal), or 3+ (positive), equivocal cases were further confirmed by in situ hybridization); T1, tumor ≤ 20 mm; T2, tumor > 20 mm and ≤ 50 mm; T3, tumor > 50 mm; T4, tumor of any size with direct extension to the chest wall and/or skin; Regional lymph node negative, no regional lymph node metastasis; positive, included N1 (mobile ipsilateral lymph node metastases, axillary levels I, II), N2 (clinically fixed or entangled lymph node metastases I, II, or ipsilateral internal mammary lymph node metastasis) or N3 (axillary level III ipsilateral lymph node metastasis with or without axillary involvement at levels I and II, or metastasis in internal mammary lymph node with level I and II axillary involvement, or supraclavicular lymph node metastasis with or without axillary or internal mammary involvement); M0, without clinical or radiographic evidence of distant metastasis; M1, distant metastasis determined clinically and radiographically and/or histologically. Hormonal receptor, positive if ER and/or PR positive; negative if both RE and RP negative.

## Data Availability

The study protocol and the datasets generated during and/or analyzed during the current study, including the identified participant data, are available upon publication from the corresponding author upon reasonable request.

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
