# Peer review of "Metabolomics by NMR Combined with Machine Learning to Predict Neoadjuvant Chemotherapy Response for Breast Cancer"

_cancers, 2022, doi:10.3390/cancers14205055_

Round 1
Reviewer 1 Report
This submission is interested while the authors should add some works into this submission: (1) some wet lab works should be added to judge the liable of their prediction; (2) The presentation can be enhanced to show their novelty; (3) The quality of the figure can be better.
Reviewer 2 Report
1. In the Materials and Methods section, a description of the group, including clinical and pathological characteristics, should be provided.
2. It is completely unclear how the response to treatment was assessed in a short period and without dividing into subgroups in accordance with at least the stage of the disease.
3. Need more information about chemotherapy regimens.
Since the group is heterogeneous, accordingly, NAC regimens differ, but nowhere in the text is the difference in response to treatment and the quality of prognosis with the same treatment regimen discussed. How many courses of chemotherapy were performed in each case, was radiation treatment performed?
4. HER2 positive patients received trastuzumab. Does this group also compare in response to others? In my opinion, this is incorrect.
5. Are the models a group in which response was assessed in combination with a metabolomic profile? But if you are talking about predicting the response to adequate NAC, then information about the hormonal and HER-2 status of the tumor will not be available unless a biopsy was performed. How do you combine this information?
Round 2
Reviewer 1 Report
It's ok for publishing now
Reviewer 2 Report
The authors seriously revised the manuscript, gave detailed answers to all the comments of the reviewer. I believe that in its present form the article can be recommended for publication.